# Multiple Occurrences of a 168-Nucleotide Deletion in SARS-CoV-2 ORF8, Unnoticed by Standard Amplicon Sequencing and Variant Calling Pipelines

**DOI:** 10.3390/v13091870

**Published:** 2021-09-18

**Authors:** David Brandt, Marina Simunovic, Tobias Busche, Markus Haak, Peter Belmann, Sebastian Jünemann, Tizian Schulz, Levin Joe Klages, Svenja Vinke, Michael Beckstette, Ehmke Pohl, Christiane Scherer, Alexander Sczyrba, Jörn Kalinowski

**Affiliations:** 1Center for Biotechnology (CeBiTec), Bielefeld University, 33615 Bielefeld, Germany; dbrandt@cebitec.uni-bielefeld.de (D.B.); msimunov@cebitec.uni-bielefeld.de (M.S.); tbusche@cebitec.uni-bielefeld.de (T.B.); mhaak@cebitec.uni-bielefeld.de (M.H.); pbelmann@cebitec.uni-bielefeld.de (P.B.); sebastian.juenemann@uni-bielefeld.de (S.J.); tischulz@cebitec.uni-bielefeld.de (T.S.); lklages@cebitec.uni-bielefeld.de (L.J.K.); vinke@cebitec.uni-bielefeld.de (S.V.); mbeckste@cebitec.uni-bielefeld.de (M.B.); asczyrba@cebitec.uni-bielefeld.de (A.S.); 2Faculty of Technology, Bielefeld University, 33615 Bielefeld, Germany; 3Department of Biosciences, Durham University, Durham DH1 3LE, UK; ehmke.pohl@durham.ac.uk; 4Evangelisches Klinikum Bethel, Institut für Laboratoriumsmedizin, Mikrobiologie und Hygiene, 33617 Bielefeld, Germany; christiane.scherer@evkb.de; 5Universitätsklinikum OWL der Universität Bielefeld, 33615 Bielefeld, Germany

**Keywords:** genomic surveillance, SARS-CoV-2, nanopore sequencing, ORF8 deletion, viral genomics

## Abstract

Genomic surveillance of the SARS-CoV-2 pandemic is crucial and mainly achieved by amplicon sequencing protocols. Overlapping tiled-amplicons are generated to establish contiguous SARS-CoV-2 genome sequences, which enable the precise resolution of infection chains and outbreaks. We investigated a SARS-CoV-2 outbreak in a local hospital and used nanopore sequencing with a modified ARTIC protocol employing 1200 bp long amplicons. We detected a long deletion of 168 nucleotides in the ORF8 gene in 76 samples from the hospital outbreak. This deletion is difficult to identify with the classical amplicon sequencing procedures since it removes two amplicon primer-binding sites. We analyzed public SARS-CoV-2 sequences and sequencing read data from ENA and identified the same deletion in over 100 genomes belonging to different lineages of SARS-CoV-2, pointing to a mutation hotspot or to positive selection. In almost all cases, the deletion was not represented in the virus genome sequence after consensus building. Additionally, further database searches point to other deletions in the ORF8 coding region that have never been reported by the standard data analysis pipelines. These findings and the fact that ORF8 is especially prone to deletions, make a clear case for the urgent necessity of public availability of the raw data for this and other large deletions that might change the physiology of the virus towards endemism.

## 1. Introduction

In December 2019, a previously unknown coronavirus causing severe cases of pneumonia started to spread from the Hubei province in China and was later termed severe acute respiratory syndrome coronavirus 2 (SARS-CoV-2) due to its relatedness to SARS-CoV [1,2]. SARS-CoV-2 rapidly reached all continents and has caused over 120 million infections and 2.6 million deaths worldwide until March 2021 [3].

Coping with increasing human immunity, improved medical treatment strategies and vaccination efforts, mutations increasing viral fitness are being positively selected. To track and monitor viral genetic diversity, viral whole-genome sequencing (WGS) is widely employed and also partly mandated by public health authorities [4]. Here, targeted tiled-amplicon sequencing, e.g., using the ARTIC-ONT protocol [5], has been proven to be a key technology for WGS especially for low viral loads samples [6].

SARS-CoV-2 ORF8 codes for an accessory protein, which is located in a hypervariable region of about 430 nt common to several SARS-similar coronaviruses and is possibly involved in immune system evasion by downregulating MHC-I as well as inhibiting Type I interferon signaling [7,8,9,10]. Several genomic deletions in ORF8, ranging from 1 to 382 nt, have been reported for SARS-CoV-2 [10,11]. It was assumed that hairpins in the ORF8 transcript region could play a role in genomic rearrangements during virus replication [11]. The so-called ∆382 variant, which was first identified in Singapore in early 2020, apparently causes infections with milder clinical symptoms [12,13]. In addition, during the SARS-CoV epidemic in 2003, a 29 nt deletion in ORF8 splitting it into ORF8a and ORF8b led to decreased viral fitness and may have reduced the severity of the epidemic [14].

Genomic surveillance of SARS-CoV-2 is of great importance, but besides identification and tracking of variants of concern (VOCs), which potentially pose the risks of higher infectivity or immune system evasion, the systematic screening for deletion mutants is also of high interest in order to monitor their spread and assess their impact on the severity of the COVID-19 disease as well as immunity in populations.

## 2. Materials and Methods

### 2.1. RNA Extraction and RT-qPCR Assays

Routine RT-PCR was performed on LightCycler 480II (Roche, Basel, Switzerland) or CFX 96 (r-biopharm, Pfungstadt, Germany) using CE-IVD labeled test kits (RIDAGENE SARS-CoV-2 (r-biopharm) Allplex SARS-CoV-2 (Seegene, Seoul, South Korea). Nucleic-acid extraction was performed utilizing semi-automated RNA extraction protocols by Maxwell RSC Blood DNA for Maxwell RSC 48 (Promega, Madison, WI, USA) or the Nucleic Acid Extraction Kit for Maelstrom 9600 (TANBead, Taoyuan City, Taiwan).

### 2.2. Nanopore Sequencing

For random-primed cDNA synthesis using the LunaRT SuperMix kit (New England Biolabs, Ipswich, MA, USA), 8 μL of extracted total nucleic acid solution were used as input. Amplicon libraries were constructed using the protocol by Freed et al. [15] utilizing the 1200 bp amplicon “midnight” primer set. Sequencing on the Oxford Nanopore GridION sequencer was conducted using R9.4.1 flow cells. LSK109 with native barcodes or RBK004 kits (Oxford Nanopore Technologies, Oxford, UK) were used for Nanopore library preparation, depending on the number of samples sequenced in a given batch.

### 2.3. Illumina Sequencing

The NEBNext ARTIC SARS-CoV-2 FS Library Prep Kit (New England Biolabs, Ipswich, MA, USA) was used for library preparation. Sequencing was performed on the Illumina MiSeq in paired-end mode (2 × 75 bp) with V3 chemistry (Illumina, San Diego, CA, USA).

### 2.4. Sanger Sequencing

Selected single-nucleotide variants (SNVs) and the ORF8 deletion were validated by Sanger sequencing. PCR products for sequencing containing the SNVs were prepared using forward and reverse primer from adjacent upstream and downstream ARTIC-V3 amplicons, respectively. The obtained approximately 1200 bp long amplicons were then prepared for sequencing using Exo-CIP (New England Biolabs, Ipswich, MA, USA) and sequenced using the inner forward and reverse primers spanning a 400 bp region. SNV regions were manually inspected using the chromatogram files in SnapGene 5.0 (GSL Biotech LLC, San Diego, CA, USA).

### 2.5. Bioinformatic and Data Analysis

Oxford Nanopore sequencing data sets were processed with the ARTIC Fieldbioinformatics pipeline (v1.2.1) according to the recommendations of the ARTIC consortium (https://artic.network/ncov-2019/ncov2019-bioinformatics-sop.html, accessed on 15 April 2021) with the following changes: sequencing reads were demultiplexed with Guppy (v.4.4.2) and kept for analysis if a barcode sequence was detected at least on one end of the sequence and the read length was within the range of 250 to 1500 nt. The process of sequencing depth normalization was modified to a coverage-based read subsampling approach to address problems of the standard amplicon-based approach with amplicon fragmentation using the RBK004 sequencing kits (see Appendix A). Modified versions of the Fieldbioinformatics tool (https://github.com/MarkusHaak/fieldbioinformatics.git, accessed on 15 August 2021) and Longshot (https://github.com/MarkusHaak/longshot.git, accessed on 15 August 2021) were deposited at Github.

Illumina data were processed using the openly available data analysis pipeline provided by the COG-UK consortium using standard parameters (https://github.com/connor-lab/ncov2019-artic-nf, accessed on 2 April 2021) [16]. The workflow is based on the ARTIC networks Fieldbioinformatics tools.

### 2.6. Phylogenetic and ORF8 Deletion Analyses

Viral genomes in GISAID (download: 3 March 2021) were scanned for the ORF8 deletion by using Virulign (v1.0.1) [17]. With Virulign, every genome was codon-correctly aligned against all ORFs of the Wuhan-Hu-1 reference genome (MN908947.3). The Virulign ORF8 nucleotides output was scanned for the deletion using a straightforward Linux grep command which resulted in nine preliminary identified deletion sequences. These sequences were further investigated by a search for SNVs that all nine sequences do have in common and are different to the Wuhan reference. We could identify eight positions in ORF1ab, ORF1a, N and S which we finally used for scanning all GISAID genomes. Thereby, we could identify 239 genomes containing the same SNV pattern.

We computed the pairwise edit distance between all sequences by using the Edlib aligner (v1.0.6) [18] with the infix mode, which does not penalize terminal gaps. Ambiguous and degenerated bases were counted as matches. Based on the distance matrix, we computed the minimum spanning tree using the Python library NetworkX (v2.5) [19]. The tree has been visualized using Cytoscape (v3.18.1) [20]. Nextstrain analysis was carried out according to the official workflow (https://github.com/nextstrain, accessed on 5 May 2021) [21,22].

### 2.7. Structural Computational Analysis of ORF8

In order to analyze the potential structural changes of the ORF8 deletion variant, the coordinates of ORF8 (PDB codes: 7JTL and 7JX6) were downloaded from the protein data bank [23]. Coordinates were manipulated and models superimposed with Coot [24]. All figures were produced with Pymol version 2.5.0 [25].

## 3. Results

### 3.1. Nanopore Long-Read Amplicon Sequencing Identified a Large Deletion in ORF8 in an Infection Chain from a Local Hospital

During the so-called second wave between 2 January and 1 March 2021, 11,041 patients, health-care workers (HCWs) and non-medical professionals from a 1250-bed campus hospital of our associated University Hospital were screened for SARS-CoV-2 using RT-PCR. During that time, 27,159 samples from the respiratory tract were examined, mainly throat swabs. Positive cases in patients or staff were handled and followed up according to national guidelines of the Robert Koch Institute. In particular, the following measures were carried out.

SARS-CoV-2 screening routines comprised RT-PCR testing of each patient either pre-inpatient or at the time of admission on the admission ward. Positively tested patients were immediately transferred to dedicated COVID-19 wards, patients with negative test results were transferred to the non-COVID-ward of the specialist clinics and were further screened once per week during inpatient treatment. HCWs in non-COVID-wards were routinely screened once a month, HCWs in the emergency department and on COVID wards once a week. In case a patient from a non-COVID ward received a positive test result during the inpatient stay, the patient was immediately transferred to the COVID ward, all other inpatients of the affected ward and all members of the staff were tested every other day for a period of 10 days. Positively tested members of the staff were released from patient care. Direct contacts were handled according to the national quarantine regime. By conducting this thorough screening routine, 394 individuals tested positive and it was possible to detect and contain 11 presumed infection clusters in the house.

Here, we report on one of those clusters, characterized by repeated transmissions between HCWs and patients, comprising 76 cases. Of these cases, 46 were subjected to amplicon WGS, which resulted in 44 complete SARS-CoV-2 genome sequences (see Appendix A). All sequences possess at least 21 SNVs compared to the Wuhan-Hu-1 reference (MN908947.3) and were assigned to Clades 20B by Nextclade (clades.nextstrain.org) and M.1 (B.1.1.294.1) by Pangolin (cov-lineages.org, accessed on 17 September 2021), respectively.

A minimum spanning tree of all 44 genomes was computed from edit distances determined by pairwise sequence comparisons, which shows five clusters of three or more viral genomes differing in at least one nucleotide position (Figure 1). Cluster A marks the beginning of the outbreak, since it contains the first identified cases (N1–N5) and has the lowest genetic distance to the Wuhan-1 reference (21 SNVs). Subsequently, infection chains continued in separate directions to Clusters B and C, respectively. Infections in Cluster B seemingly were contained, whereas virus strains from Cluster C formed Clusters D and E. Interestingly, Ward G, depicted in orange and mainly comprising Clusters D and E, is located at another facility (Figure 1).

Upon further investigation, we observed a distinct region in the ORF8 gene without any read coverage in all of these outbreak-related samples. The missing region spans 168 nucleotides from Positions 28,003 to 28,170 in the SARS-CoV-2 genome and could not be attributed to the dropout of a full-length amplicon. Manual investigation of sequencing reads proved it to be an actual in-frame genomic deletion. This was additionally validated by Sanger sequencing in all samples and is further referred to as the ∆168 variant. Furthermore, we detected this deletion in 30 further outbreak-related samples, which were not sequenced, via PCR and agarose gel electrophoresis.

In the ∆168 variant, an in-frame deletion led to amino acids 38 to 93 being removed and residue 37 changed from cysteine to serine in the ORF8 protein, resulting in a truncated 65 amino acid variant. The wildtype ORF8 auxiliary protein consists of 121 amino acids comprising an N-terminal signal sequence for import into the endoplasmatic reticulum (ER) followed by an immunoglobulin (Ig) domain. This domain is likely to interact with a variety of host proteins involved in the immune response [8,9]. The crystal structure of ORF8 shows that the protein adopts the expected Ig-fold of a β-sandwich consisting of two β-sheets [23]. Each monomer is stabilized by three intramolecular disulphide bridges (Cys25–Cys90, Cys37–Cys102, Cys61–Cys81) (Appendix A). The protein forms a tightly packed homodimer with a dimerization interface stabilized by an intermolecular disulfide bond (Cys20–20) and involving the ORF8 unique _73_YIDI_76_ motif [26].

The deletion reported here removes not only the ORF8 unique insertion, which represents the major difference between ORF8 and ORF7 (amino acids 48–77), but four of the eight β-sheets and all of the intramolecular disulphide bridges (Appendix A). Although the disulphide bridges in full Ig domains are not necessarily essential for proper folding [27] it is unlikely that this truncated form alone forms a fully folded and functional protein. This notion is consistent with the observation that SARS-CoV-2 missing the ORF8 gene has been suggested to be infectious albeit associated with milder symptoms.

Inspection of the NGS raw data and Sanger sequencing results showed that these were homogenous viral populations. However, after data processing using the ARTIC software pipeline, the deletion had not been detected.

The ARTIC V3 primer set, which consists of 218 primers for a total of 98 amplicons, (github.com/artic-network/artic-ncov2019/blob/master/primer_schemes/nCoV-2019/V3/, accessed on 3 April 2021), is used by many academic groups and medical laboratories for SARS-CoV-2 amplicon sequencing, as it was one of the first published whole-genome amplicon sets for SARS-CoV-2 and is now included in several commercial SARS-CoV-2 sequencing kits. The primers “nCoV-2019_93_LEFT” and “nCoV-2019_92_RIGHT” bind within the deleted region, leading to the dropout of Amplicons 92 and 93 (Figure 2). Here, employing a modified sequencing approach, using 29 amplicons of 1200 bp each [15], we obtained reads spanning the deletion, because no primer-binding sites were affected by the deletion.

Since we did not observe the deletion in the final output of the ARTIC data analysis pipeline, we aimed to reevaluate our data analysis strategy. The ARTIC pipeline offers two workflows for variant calling from nanopore amplicon sequencing data. One uses medaka-variant (github.com/nanoporetech/medaka, accessed on 15 May 2021) for initial variant calling followed by longshot [28] for variant filtering and the other one only uses nanopolish [29] for variant calling from raw signal-level data.

We discovered that in the medaka workflow the ∆168 deletion was initially detected by medaka-variant, but, by design, filtered out during the longshot step. This is caused by two separate filter settings: one is the “max_cigar_length” option passed to longshot, which is per default set to 20 nt and the other is a hard-coded upper boundary removing all indels > 50 nt (see Appendix A for detailed information). Since the initially detected ∆168 deletion is consequently removed from the final variant calling result, this leads to the omission of the ∆168 deletion from the final consensus sequence, as the wildtype sequence is kept in place during consensus building. The nanopolish pipeline, however, was previously shown to be able to reliably detect a 34 nt long deletion in SARS-CoV-2 genomes [6].

When analyzing Illumina short-read data of ∆168 variant samples using the ARTIC-V3 amplicons and the COG-UK pipeline [16], we occasionally observed correctly assembled ∆168 genomes, such where the deletion is marked by a respective stretch of ambiguous bases. Most probably with sufficiently high read coverage, fragments of longer amplicons resulting from primer-hopping are sequenced and enable the calling of the ORF8 deletion although the respective 400 bp PCR products could not be amplified (Figure 2).

### 3.2. The Same 168 Bases Long Deletion Has Occurred Independently throughout Europe

For a systematic study on whether this deletion occurs in other SARS-CoV-2 genome sequences, we pursued different strategies. As mentioned above, this included searching for correctly assembled ∆168 genomes, that have the same or a very similar mutation pattern but where the deletion is either covered by the sequence from the reference or masked by a respective stretch of ambiguous bases.

Firstly, we found nine correctly assembled ∆168 variant genomes in the GISAID database (see Appendix A) with the earliest ∆168 variant sequenced in Italy in June 2020 (EPI_ISL_636465). Phylogenetic analysis revealed that the ∆168 variant sequences from GISAID, similar to our sequences, belong to Clade 20B.

Secondly, we determined a common set of eight SNVs in protein-coding regions present in the nine preliminary ∆168 variant genomes from GISAID (see Appendix A). Interestingly, these SNVs are also present in all outbreak samples, when allowing ambiguous bases to be counted as matches. In the GISAID database, we found 239 sequences also containing this conserved set of eight SNVs.

Thirdly, we extended our search for GISAID sequences containing 168 ambiguous bases at genome Positions 28,003 to 28,170, which identified 13 additional genome sequences with the ∆168 ORF8 deletion. All but one of those virus strains belong to Clade 20B. The only exception is the sequence EPI_ISL_664905 sampled in November 2020 in the UK, that belongs to Clade 20E (EU1).

By using the GISAID sequence accessions, the European Nucleotide Archive (ENA) was screened for the corresponding raw read data and 97 of these data sets were identified (see Appendix A). By searching for identical matches of an 18 base sequence region spanning the ∆168 deletion in these short-read Illumina data sets, hits for 87 datasets were reported (see Appendix A). We, therefore, assume that samples with the common 8 SNVs and those with stretches of ambiguous bases also, at least partly, contain the same deletion, which is masked by flaws in the analysis pipelines. Additionally, we determined tree branch lengths from sample collection dates (Appendix A). Forty-two of the raw data sets showing the ∆168 deletion belong to a cluster of 51 sequences from the UK sampled between October 2020 and January 2021 (Figure 3 and Appendix A). The corresponding consensus sequences from GISAID contain up to 591 ambiguous bases in the ORF8 region (Genomic Positions 27,800 to 28,574), where sequences originating from a higher sequencing depth tend to have shorter stretches of ambiguous bases.

Phylogenetic analysis also reveals that the direct predecessors of the ∆168 cluster stem from Denmark (Figure 3). However, the genomes do not show the ∆168 deletion and corresponding raw data were not available. Since in these cases nanopore sequencing with long amplicons of 1000 to 1500 bp and medaka-variant for consensus calling was employed (personal communication, Thomas Yssing Michaelsen, 1 June 2021), it can be assumed that the deletion was not called due to the technical setup.

After the end of the hospital outbreak, seven additional SARS-CoV-2 genome sequences containing the conserved set of 8 SNVs and the ∆168 deletion covered by variable stretches of ambiguous bases were submitted to GISAID (see Appendix A). The submitting laboratory kindly provided us with the raw data of three of those sequences, enabling us to correct ill-defined deletion borders after reanalysis and manual inspection. However, the exact positioning of the ∆168 deletion did not influence the phylogenetic analyses.

Sampling locations of all seven sequences also point to a local spread from the hospital, as all sequences originate from North-Rhine Westphalia and especially the Bielefeld region in eastern Westphalia. SNV patterns and the phylogenetic analysis indicates that there have been several transmission events from the hospital outbreak to the public. One event seems to have occurred from Cluster B to NW-RKI-I-026594, NW-RKI-I-029752, NW-RKI-I-036765 and NW-RKI-I-013019, while the sequences NW-RKI-I-019280 and NW-RKI-I-019218 seem to stem from Cluster A and NW-RKI-I-034291 is identical to most sequences of Cluster C (Figure 4).

### 3.3. Standard Amplicon Sequencing and Variant Calling Pipelines Potentially Hide Other Large Deletions

We wondered whether missed large deletions could be a widespread problem with SARS-CoV-2 amplicon sequencing and aimed to investigate the GISAID database for further appearances of other large ORF8 deletions. We based our analysis on a MAFFT alignment [31] of 830,039 GISAID sequences downloaded on 3 March 2021 (also see Appendix A). As illustrated above, deletions are either represented by stretches ambiguous bases or by gaps in the consensus sequence, dependent on sequencing technology and analysis pipeline. We, therefore, extracted and visualized the lengths of stretches of ambiguous bases and gaps in the multiple sequence alignment in a 1000 bp region surrounding the ORF8 gene (Figure 5).

Peak occurrences of continuous ambiguous base stretches of lengths around 280 and 590 bp correspond to a single or two missing amplicons of the ARTIC-V3 primer scheme, respectively, which was clearly deducible from the underlying genome sequences. The peak at around 140 bp is essentially also indicating an amplicon dropout, but with an artificial non-N stretch right in the middle of a 280 bp long region.

We, furthermore, identified a prominent bin with stretches of ambiguous bases of 220 to 230 bases (Figure 5), which seemingly partly contains another unknown ORF8 deletion of 225 bases ranging from Position 28,003 to 28,227 (data not shown). However, this particular bin also contains stretches of ambiguous bases that correspond to the dropout of a single amplicon, which are shorter than 280 bp because of variations in the data analysis pipelines. Nonetheless, these findings indicate that there are a variety of deletions in the ORF8 region never identified by the standard data analysis pipelines.

## 4. Discussion

By tiled-amplicon genome sequencing, we were able to elucidate an outbreak event in a local hospital, which was difficult to contain over the course of over two months. Most outbreak infections were only identified by routine contact screening, due to generally rather mild symptoms. This shows that genomic surveillance of SARS-CoV-2 and infectious agents, in general, can serve an important role in the elucidation of infection chains and outbreak events. To achieve this, thorough routine screenings of patients and HCWs, as well as contact tracing is needed.

Long-read sequencing data showed a defined 168 nt deletion in ORF8 in the outbreak samples. This finding is of special interest, since during the SARS-CoV epidemic in 2003/2004 a 29-nt deletion in ORF8, essentially splitting ORF8 into two coding regions, is discussed as having accelerated the end of the outbreak [14].

In early 2020, a 382 nt deletion comprising the complete SARS-CoV-2 ORF8 and the 3′-end of ORF7 has been discovered in Singapore [12]. Clinical data have shown that infections caused by the ∆382 variant led to milder symptoms including milder hypoxic conditions and lower cytokine activity than infections by the SARS-CoV-2 wildtype [13], which may be due to the ORF8 gene product functioning as an inhibitor of MHC-I and Type I interferon signaling [8,9]. Interestingly, the B.1.1.7 variant of concern contains a nonsense mutation Q27* in ORF8 [32], but here, an attenuated severity of disease was not reported.

In this study, we discovered that sequencing approaches based on the widely employed ARTIC-V3 primer set are only able to call the ∆168 variant with an extremely high sequencing depth that allows detection of primer-hopping amplicons spanning the deletion. With nanopore sequencing of the V3 amplicons, detection of the ∆168 variant, however, would be prevented by stringent read length filtering before variant calling. Additionally, the ARTIC medaka pipeline per default removes SNVs larger than 20 bases, consequently including the ∆168 variant. The initial observation of the ∆168 variant was therefore dependent upon manual inspection of the raw data. In the future, longer amplicons should be routinely used, and the ARTIC medaka software pipeline should be revised to enhance its suitability for the detection of large deletions. Soon to be released ARTIC Fieldbioinformatics version 1.3 no longer employs longshot in the medaka workflow and should therefore no longer cause the issue of omitted large deletions. Nonetheless, only the deposition of unfiltered, raw sequencing reads allows the scientific community to assess the spread of deletion strains.

We applied several different search strategies for the occurrence of the ∆168 variant in the GISAID database, leading to the discovery of over 100 virus genome sequences (see Appendix A) with the exact same deletion across Europe occurring as early as June 2020. Interestingly, almost all ∆168 variant genomes stem from Europe and nearly all belong to Clade 20B. On its own this would suggest that the deletion happened once and subsequently spread across Europe, accumulating further mutations over time. However, phylogenetic analysis suggests that the deletion emerged independently on several occasions especially because of the fact that one of the ∆168 variant genomes belongs to Clade 20E, differing in five positions from Clade 20B. Further, given the assumption that there has been a single deletion event, SNVs in the 20B strains do not fit the presumed accumulation over the course of the pandemic, as SNVs being found in early samples are not present in later genome sequences. It remains to be elucidated by more in-depth analyses whether this deletion marks a molecular hotspot or the deletion itself is under positive selection.

We found seven additional occurrences of the ∆168 variant by screening GISAID data, after the hospital outbreak ended. SNV patterns matched the outbreak sequences, showing that the ∆168 variant has already spread to the public in North-Rhine Westphalia, Germany. Virus strains possibly associated with milder clinical courses tend to be underrepresented in genome databases, because infections are less likely to be identified, and sequencing efforts in Germany are focused on virus samples suspected to contain so-called VOCs mainly characterized by several spike protein mutations. Consequently, large genomic deletions and their influence on the course of the SARS-CoV-2 pandemic are probably under-investigated, although they could play a key role in the progression of COVID-19 into an endemic disease.

## Figures and Tables

**Figure 1 viruses-13-01870-f001:**
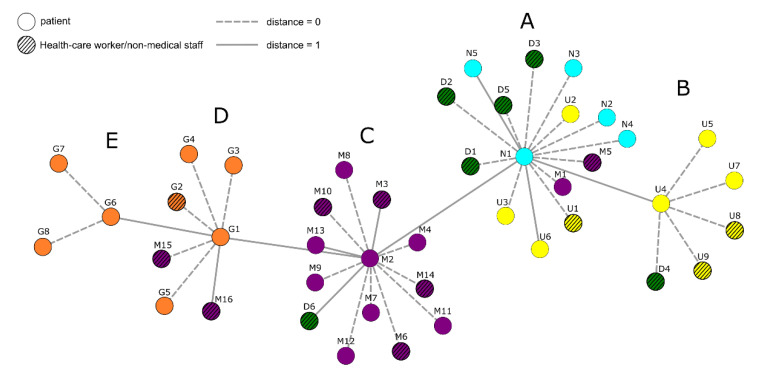
Genomic surveillance of SARS-CoV-2 hospital outbreak with the ∆168 variant across multiple wards. Genetic distances were determined by computing the minimum spanning tree from pairwise edit distances between 44 complete SARS-CoV-2 sequences stemming from the hospital outbreak. Edit distances of zero or one are drawn by dashed or solid lines, respectively. Virus genomes isolated from patients and HCWs/non-medical staff are displayed as circles and were assigned a color corresponding to the clinical ward. HCWs/non-medical staff are shown with a broadly outlined circle and black crosslines, patients are shown with a thin outline. For each of the clusters of closely related sequences **A**, **B**, **C**, **D** and **E**, samples taken earliest are displayed at the center. Distances were calculated using the Edlib aligner [18], the minimum-spanning tree was computed using the Python library NetworkX [19] and visualized using Cytoscape [20].

**Figure 2 viruses-13-01870-f002:**
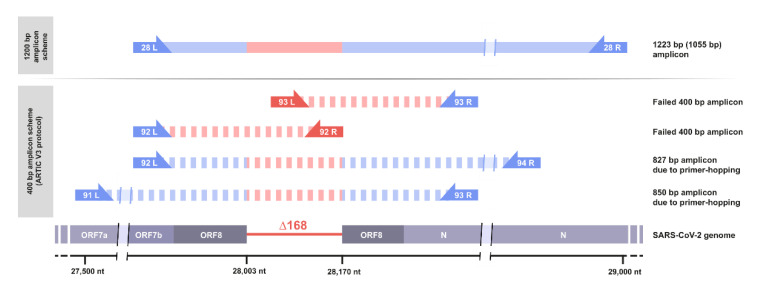
Schematic view on the ORF8 gene region and the respective amplicons using different amplification protocols. Amplicon 28 of the amplicon scheme by Freed et al. [15], using 1200 bp amplicons, is shortened to 1032 bp in the ∆168 variant (top). Amplicons 92 and 93 of the ARTIC-V3 amplicon scheme fail due to the deletion of primer-binding sites. However, more distant primer-pairs may interact and lead to longer amplicons spanning the ∆168 deletion, thus enabling sequencing using the ARTIC-V3 protocol.

**Figure 3 viruses-13-01870-f003:**
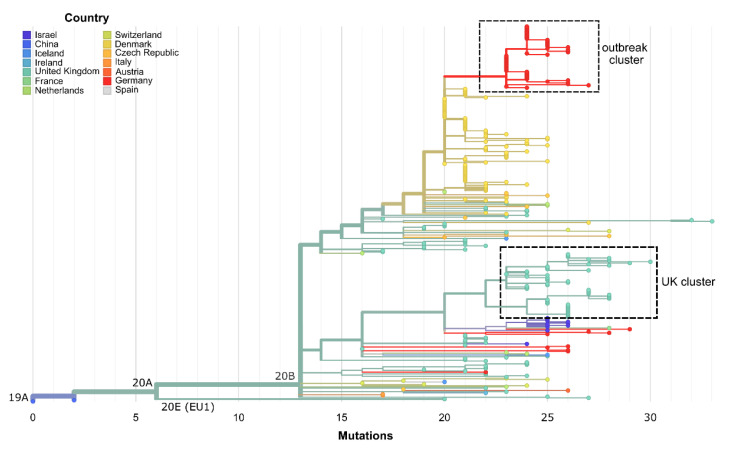
Maximum-likelihood phylogenetic tree from Nextstrain comprising 307 SARS-CoV-2 genome sequences [22,30]. Branch lengths are determined by genetic divergence. The dataset consists of nine genomes from GISAID with the ∆168 ORF8 deletion, 239 genomes with a similar SNV signature, 13 genomes with the ∆168 ORF8 deletion masked by a stretch of Ns, 44 genomes from this study stemming from the hospital outbreak and two Wuhan reference sequences (WH01, Hu-1). The hospital outbreak cluster is visible at the top (red) and is directly preceded by sequences from Denmark (yellow), which have similar SNV signatures but no large deletion in ORF8. A distinct cluster of sequences from the UK (turquoise) also shows the deletion in the raw sequence data, whereas the consensus sequences contain variable stretches of ambiguous bases. Seven sequences listed in Appendix A have been removed due to missing metadata by the Nextstrain tree filter.

**Figure 4 viruses-13-01870-f004:**
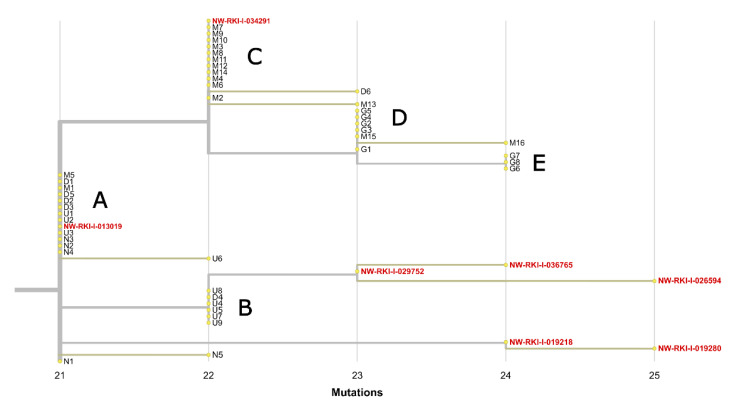
Maximum-likelihood phylogenetic tree from Nextstrain [22,30] comprised of the 44 complete SARS-CoV-2 genomes from the hospital outbreak samples and seven SARS-CoV-2 sequences most probably resulting from a local spread of the hospital outbreak to the public. Five infection clusters (A, B, C, D, E) are clearly delineated from each other and outbreak leakage (red) obviously occurred from Clusters A, B and C.

**Figure 5 viruses-13-01870-f005:**
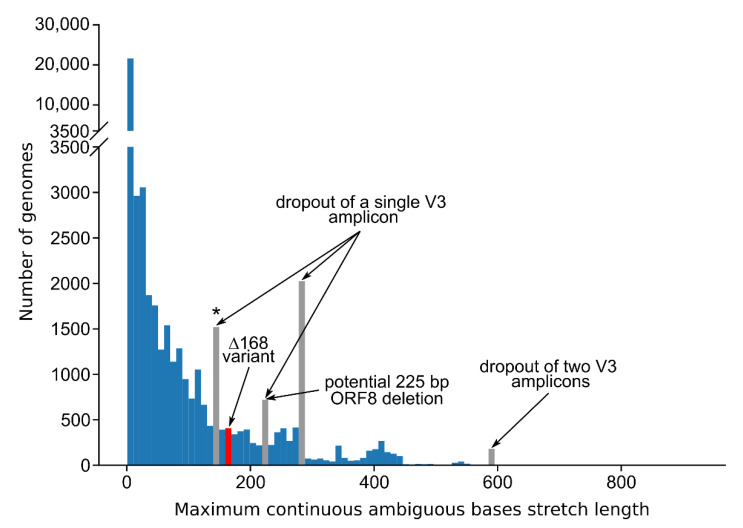
Histogram of the lengths of stretches of ambiguous bases and gaps in the ORF8 gene region in SARS-CoV-2 sequences from GISAID. Positions 27,577 to 28,577 were extracted from a MAFFT alignment [31] of 830,039 sequences downloaded from GISAID on 3 March 2021. The bin size was fixed at 10 bp. For each sequence the length of the longest continuous ambiguous base stretch or gap was determined. Only stretches starting and ending within the selected alignment region were counted. Stretches that are most probably attributable to certain amplicons from the ARTIC-V3 scheme that are completely missing are colored grey. The bin around 220–230 bp possibly contains a novel 225 bases long ORF8 deletion variant (28,003 to 28,227) in addition to stretches of ambiguous bases corresponding to the dropout of a single amplicon. ARTIC-V3 Amplicons 92 and 93 are fully encompassed by the selected alignment region while Amplicons 91 and 94 are only partly included. The grey bin marked with an asterisk (140 to 150 bp) corresponds to the dropout of Amplicon 91. The bin containing the ∆168 variant is colored red.

## Data Availability

Raw sequencing data is available from the European Nucleotide Archive under Project Accession Number PRJEB45912.

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
