# Peer review of "Multiple Occurrences of a 168-Nucleotide Deletion in SARS-CoV-2 ORF8, Unnoticed by Standard Amplicon Sequencing and Variant Calling Pipelines"

_viruses, 2021, doi:10.3390/v13091870_

Round 1

Reviewer 1 Report

This paper presents an analysis that identifies a 168-nucleotide long deletion in ORF8 of SARS-CoV-2, the virus causing the ongoing COVID-19 pandemic, in both a local outbreak in Germany as well as in other European samples submitted to the GISAID database. This is an important contribution to the literature dealing with sequence variation in SARS-CoV-2, and suggests that a substantial part of this variation may be missed by existing analysis tools. The paper is well-written and convincingly argued; I have a few relatively minor comments that I would like to see the authors address as I detail below.

Major comments.

1) It would be very interesting to discuss in more detail the functional consequences of the identified deletion, and at least speculatively propose a possible explanation for the milder clinical outcomes. The authors mention the complete deletion of ORF8 identified in Singapore and the hypothesis that the gene product may inhibit MHC-I and type 1 interferon signalling; that being said, a complete deletion has a very different effect than a partial one (the fact that this deletion is in-frame means only a few codons are omitted).

2) Along those lines, it may be interesting to attempt to reconstruct the protein structure resulting from the wild-type and the partially deleted version of this ORF (especially if the region is believed to be a coding region). This may be possible to do eg with the recently released AlphaFold software.

3) The detection of the deletions appears to be a consequence of two factors: the specific long amplicon technology used by the authors and their bespoke modification of the standard analysis pipeline for the resulting data. In order to ensure complete reproducibility of the latter it would be important to create a downloadable and installable version of their modified pipeline, not just a pull request, to the extent that this is made possible by the original pipeline's software license.

Minor comments.

1) The authors do not mention whether the tree in Figure 2 (as well as the trees in the Supplementary Data) has been obtained purely from the sequence alignment data, or has also taken into account the collection date (which, presumably, is closely correlated with the infection date); only the latter would allow reliable conclusions to be drawn about the order in which transmission may have happened (tools such as TransPhylo, for instance, may be helpful in trying to convert a phylogeny into a transmission tree when enough dates are known and the sampling is dense, as was the case here).

2) It was not clear that the ambiguous bases appearing in the corresponding positions of ORF8 are necessarily indicative of a deletion in that region; it will be helpful for the authors to clarify this somewhat more in detail. 

3) Generally speaking, I would recommend that (short of carrying out a more comprehensive analysis of the mechanistic consequences of the deletion) the authors put more emphasis on the fact that their methods succeed where commonly used pipelines fail to identify long deletions, and that they have been able to identify samples in a number of European countries where the same or similar deletions have occurred; this might also be an opportunity for a stronger call for the submitting laboratories to upload the full raw data, rather than the processed data, from their sequencing experiments in order to strengthen the data's usability for global surveillance.

Author Response

Major comments.

(please see our answers in red)

1) It would be very interesting to discuss in more detail the functional consequences of the identified deletion, and at least speculatively propose a possible explanation for the milder clinical outcomes. The authors mention the complete deletion of ORF8 identified in Singapore and the hypothesis that the gene product may inhibit MHC-I and type 1 interferon signalling; that being said, a complete deletion has a very different effect than a partial one (the fact that this deletion is in-frame means only a few codons are omitted).

Answer: We are grateful for the suggestion to analyse the consequences of the deletion in greater detail. Therefore, we consulted an expert in protein structure and folding from the University of Durham (Prof. Ehmke Pohl) and included him as a co-author. Based on the available crystal structure we now have extended our analysis (lines 187 ff.). Prof. Pohl additionally provided analyses on the structures of wildtype ORF8 and the ∆168 deletion variant. Supplemental Figure S1 has been introduced, which shows a comparison between both protein structures. The deletion removes four of eight beta sheets, three intramolecular disulphide bridges and the ORF8 unique 73YIDI76 motif, which enables dimerization of ORF8. We therefore consider the deletion to strongly impair folding and functionality of the ORF8 protein. We are therefore confident that the functional consequences of the ∆168 deletion may be the same as with the complete deletion of ORF8 identified in Singapore.

2) Along those lines, it may be interesting to attempt to reconstruct the protein structure resulting from the wild-type and the partially deleted version of this ORF (especially if the region is believed to be a coding region). This may be possible to do eg with the recently released AlphaFold software.

Answer: Unfortunately, the structure prediction pipeline Alphafold can not be considered an appropriate tool as the crystal structure had already been determined independently by two groups. Nevertheless, as suggested we ran the truncated sequence through Alphafold. As expected, the program models the remaining two C-terminal strands quite well, which is not surprising given a 100% sequence identity. The N-terminal part, however, is predicted to adopt a short alpha helix, which seems very unlikely, and mainly random coil to maintain a compact protein fold. Even if this model is correct, it is highly unlikely to fulfil the same or (partial) biological function as the full-length protein. As any validation of the predicted model by X-ray crystallography or NMR is clearly out of the scope of this paper we would like to refrain from a purely speculative discussion.

3) The detection of the deletions appears to be a consequence of two factors: the specific long amplicon technology used by the authors and their bespoke modification of the standard analysis pipeline for the resulting data. In order to ensure complete reproducibility of the latter it would be important to create a downloadable and installable version of their modified pipeline, not just a pull request, to the extent that this is made possible by the original pipeline's software license.

Answer: We deposited our sequence analysis pipeline, including our changes to the coverage-based sequencing depth normalization, in the following Github repository (https://github.com/MarkusHaak/fieldbioinformatics.git) Further, we deposited a modified version of longshot (https://github.com/MarkusHaak/longshot.git), which prevents the exclusion of >50 bp indels. This needs to be manually compiled and linked after installing fieldbioinformatics. Notably, the latest developer version of ARTIC fieldbioinformatics (v1.3-dev) no longer includes longshot and is solely based on Medaka version 1.2.3 for variant calling. We also included this information in the discussion section (lines 393-395). Starting from line 225, we added some additional information on the flawed pipeline settings that lead to the inability of the ARTIC pipeline to identify large deletions.

Minor comments.

1) The authors do not mention whether the tree in Figure 2 (as well as the trees in the Supplementary Data) has been obtained purely from the sequence alignment data, or has also taken into account the collection date (which, presumably, is closely correlated with the infection date); only the latter would allow reliable conclusions to be drawn about the order in which transmission may have happened (tools such as TransPhylo, for instance, may be helpful in trying to convert a phylogeny into a transmission tree when enough dates are known and the sampling is dense, as was the case here).

Answer: The reviewer probably refers to Figure 3, where the tree has only been obtained from the sequence alignment data (Nextstrain view option: “Divergence”). However, Nextstrain also allows the branch lengths to be determined by the sample collection dates. Employing this option we added Supplementary Figure S3, which shows that both view options are closely correlated. Although a real transmission tree would be useful, we refrained from creating such, because of the limited number of sequences in Figure 3 where we could prove the presence of the deletion from the raw data.

2) It was not clear that the ambiguous bases appearing in the corresponding positions of ORF8 are necessarily indicative of a deletion in that region; it will be helpful for the authors to clarify this somewhat more in detail. 

Answer: We thank the reviewer for bringing up this issue. Ambiguous bases are indeed not in all cases indicating a deletion in a given region, as the dropout of one (or more) amplicon(s) may lead to the same result. We therefore aimed to use our wording carefully, as we could only be sure about the presence (or absence) of the deletion, when raw data was available. Supplementary Figure S2 shows all sequences where we were able to obtain the underlying raw data and to confirm the deletion. If the reviewer would like us to revise a certain text passage that is unclear, we would be happy to do so.

3) Generally speaking, I would recommend that (short of carrying out a more comprehensive analysis of the mechanistic consequences of the deletion) the authors put more emphasis on the fact that their methods succeed where commonly used pipelines fail to identify long deletions, and that they have been able to identify samples in a number of European countries where the same or similar deletions have occurred; this might also be an opportunity for a stronger call for the submitting laboratories to upload the full raw data, rather than the processed data, from their sequencing experiments in order to strengthen the data's usability for global surveillance.

Answer: We have modified the abstract to make a more pronounced statement, but due to the size limitation only by a few words. It now reads „These findings and the fact that ORF8 is especially prone to deletions, make a clear case for the urgent necessity of public availability of the raw data for this and other large deletions that might change the physiology of the virus towards endemism.“ (additions in boldface).

Reviewer 2 Report

This paper is generally well-written, but the two key claims have problems. The first claim, i.e., the finding of a set of viral genomic sequences with a deletion, is not accompanied with a convincing argument of significance. Deletion in the viral genome is not rare in coronaviruses. In fact, it happens so very frequently to give us huge headaches in aligning the viral genomic or gene sequences. For this reason, finding a particular deletion by itself is not significant. However, there are ways in which deletions or any other mutations are significant. For example, if all those patients infected with the deletion variant share certain symptoms, i.e., there is association between genotype and phenotype. Even if we do not understand the cause-effect relationship between the viral deletion and the patient symptoms, such association between the two would be interesting. However, the authors hardly spent any effort to highlight the significance of the deletion.

The second claim made by the authors is that the particular deletion is missed by a certain sequencing/assembly protocol. If this is well established, and if the sequencing/assembly protocol is frequently used, then many published viral genomes would be wrong in an important way. However, I do not think that this claim is well established. Obtaining correct sequence depends on both complete sequence coverage and correct sequence assembly. In genomic sequences with many repeats, even if all sequences have been covered multiple times, the assembly is made difficult because the two ends of a sequence fragment could match the end of tens or even hundreds of other sequences. However, viral genome assembly is simple because 1) there are hardly any repeats and 2) the genome is small is exhaustive search of the best assembly is possible. 

The authors stated that "However, after data processing using the ARTIC software pipeline, the deletion had not been detected." In other words, the missing deletion in the assembled genome is due to problematic sequence assembly. In particularly, the authors claim that "We discovered that in the medaka workflow, by design, longshot filters out all indels >20 bases, leading to the omission of the Δ168 deletion in the final consensus sequence by replacing this region with the reference sequence (see Supplementary Text for detailed information)." This is not a substantiated claim. What is the evidence for "replacing this region with the reference sequence"? If the authors do have the evidence, don't hide it in supplementary text because this is in fact the only selling point in the paper.

There are some other minor issues. For example, SNV is never defined. It should be defined in its first appearance in the text.

Author Response

Comments and Suggestions for Authors

(please see our answers in red)

This paper is generally well-written, but the two key claims have problems. The first claim, i.e., the finding of a set of viral genomic sequences with a deletion, is not accompanied with a convincing argument of significance. Deletion in the viral genome is not rare in coronaviruses. In fact, it happens so very frequently to give us huge headaches in aligning the viral genomic or gene sequences. For this reason, finding a particular deletion by itself is not significant. However, there are ways in which deletions or any other mutations are significant. For example, if all those patients infected with the deletion variant share certain symptoms, i.e., there is association between genotype and phenotype. Even if we do not understand the cause-effect relationship between the viral deletion and the patient symptoms, such association between the two would be interesting. However, the authors hardly spent any effort to highlight the significance of the deletion.

Answer: We thank the reviewer for bringing up this very important issue. We agree that genomic deletions in Coronaviruses are not rare, which can also be deduced from Figure 5. However, most deletions span a range of 1-10 nucleotides and larger deletions are increasingly rare.
Unfortunately, our study was not set up as an observational cohort study. We became aware of the ORF8 deletion retrospectively and it was hard to follow-up patients symptoms and disease outcomes. Nonetheless, to highlight the significance of the deletion, we consulted an expert in protein structure and folding from the University of Durham (Prof. Ehmke Pohl). He provided analyses (
line 187 ff.) on the structures of wildtype ORF8 and the ∆168 deletion variant and was added to the list of authors. We introduced Supplementary Figure S1, which shows a comparison between both protein structures. The deletion removes four of eight beta sheets, three intramolecular disulphide bridges and the ORF8 unique 73YIDI76 motif, which enables dimerization of ORF8. We therefore confidently consider the deletion to strongly impair folding and functionality of the ORF8 protein. This allows us to assume that the clinical data obtained by Su et al. (2020), who observed a complete deletion of ORF8, is also relevant in our case.

The second claim made by the authors is that the particular deletion is missed by a certain sequencing/assembly protocol. If this is well established, and if the sequencing/assembly protocol is frequently used, then many published viral genomes would be wrong in an important way. However, I do not think that this claim is well established. Obtaining correct sequence depends on both complete sequence coverage and correct sequence assembly. In genomic sequences with many repeats, even if all sequences have been covered multiple times, the assembly is made difficult because the two ends of a sequence fragment could match the end of tens or even hundreds of other sequences. However, viral genome assembly is simple because 1) there are hardly any repeats and 2) the genome is small is exhaustive search of the best assembly is possible. 
The authors stated that "However, after data processing using the ARTIC software pipeline, the deletion had not been detected." In other words, the missing deletion in the assembled genome is due to problematic sequence assembly. In particularly, the authors claim that "We discovered that in the medaka workflow, by design, longshot filters out all indels >20 bases, leading to the omission of the
Δ168 deletion in the final consensus sequence by replacing this region with the reference sequence (see Supplementary Text for detailed information)." This is not a substantiated claim. What is the evidence for "replacing this region with the reference sequence"? If the authors do have the evidence, don't hide it in supplementary text because this is in fact the only selling point in the paper.

Answer: We very much concur with the reviewer that viral genome assembly is (normally) not a difficult bioinformatic problem. As sequencing is done using tiled-amplicons, de novo assembly has proven to be rather ineffective in our own hands, probably due to small overlaps between amplicons. With the ARTIC software pipeline, viral mutational analysis is based on read mapping and subsequent variant calling using either nanopolish or medaka (+longshot). The main steps are:
1. Indexing the ref & aligning the reads with minimap2,
2. Variant calling on each read group, either using the medaka or nanopolish workflow,
3. Checking and filtering the VCFs.
We discovered that large deletions are missed due to faulty settings during step 3. In the medaka version of the ARTIC pipeline, “medaka” is used for initial variant calling, whereas “longshot” filters the detected variants. The
∆168 deletion is initally reliably identified by “medaka”, however, there are two parameters concerning the “longshot” tool that cause the removal of the already identified deletion. The first one is the “max_cigar_indel” parameter, that is set to 20 bp from the ARTIC fieldbioinformatics pipeline upon “longshot” execution, which filters all alignments containing indels longer than 20 bp in the CIGAR string. The second one is a hard-coded boundary of 50 bp, that is applied to all potential variants that are handed to “longshot” by other tools (in this case “medaka”). Starting from line 225, we added some additional information on the pipeline settings that lead to the inability of the ARTIC pipeline to identify large deletions.

As the modification of said parameters (software deposited here: https://github.com/MarkusHaak/fieldbioinformatics.git; https://github.com/MarkusHaak/longshot.git) has led to the successful identification of the ORF8 deletion, we are sure that we identified the correct bottlenecks.
Notably, the latest developer version of ARTIC fieldbioinformatics (v1.3-dev) no longer includes longshot and is solely based on Medaka version 1.2.3 for variant calling. It can therefore be assumed that this issue will be resolved in the near future, although it surely affects many datasets that are already deposited in the databases.
We also included this information in the discussion section (lines 393-395).

There are some other minor issues. For example, SNV is never defined. It should be defined in its first appearance in the text.

Answer: For clarification, we added the definition of the term “SNV” in line 81 .

Reviewer 3 Report

This manuscript described the detection of the ORF8 deletion in SARS-CoV-2, which was reported to result in milder infections. Since this deletion has also been reported reducing the severity of the epidemic in SARS-CoV in 2003, genomic surveillance of SARS-Cov-2 mutants with this deletion should be constantly monitored, which may provide clues of how they may impact on the disease severity. They further examined SARS-CoV-2 genomes from GISAID and reported that these variants were present in other European countries. 

They found truncated a 65-aa variant comparing to 121-aa from wildtype, which is an N-terminal signal sequence for import into ER followed by 
an immunoglobulin domain. An alteration of folded ORF8 protein structure has changed how molecules were packed, which is suggested 
by the authors to associate with milder symptoms for these variants. 

In the effort to obtain the genomes for these variants, they reported a modification of existing ARTIC pipeline which was not able detect the deletion in a first place. 

There are two places (line#215, #241) that Figure 1 was mistakenly
replaced by Figure 2. In line#318, Figure 3 should be used instead of Figure 4. In line#337, Figure 4 should be used instead of Figure 5. 

Overall the data gathering and sequencing methods used in this manuscript are well handled/implemented. The description of research outcomes were well written. Discussion of various methods and approaches in genome sequencing were excellent. 

Author Response

This manuscript described the detection of the ORF8 deletion in SARS-CoV-2, which was reported to result in milder infections. Since this deletion has also been reported reducing the severity of the epidemic in SARS-CoV in 2003, genomic surveillance of SARS-Cov-2 mutants with this deletion should be constantly monitored, which may provide clues of how they may impact on the disease severity. They further examined SARS-CoV-2 genomes from GISAID and reported that these variants were present in other European countries. 

They found truncated a 65-aa variant comparing to 121-aa from wildtype, which is an N-terminal signal sequence for import into ER followed by 
an immunoglobulin domain. An alteration of folded ORF8 protein structure has changed how molecules were packed, which is suggested 
by the authors to associate with milder symptoms for these variants. 

In the effort to obtain the genomes for these variants, they reported a modification of existing ARTIC pipeline which was not able detect the deletion in a first place. 

There are two places (line#215, #241) that Figure 1 was mistakenly
replaced by Figure 2. In line#318, Figure 3 should be used instead of Figure 4. In line#337, Figure 4 should be used instead of Figure 5. 

Our reply: We thank the reviewer for critically reading our paper and pointing out these errors. We think that these problems arose, because figure numbering seems to be broken in the PDF version of the manuscript. Both Figures 1 and 2 are labeled as “Figure 1” and consequently the numbering of the subsequent figures has changed as well. The figure references in the manuscript body, however, are still correct. We will upload a revised version of the manuscript, where this error (probably due to MS Word formatting) hopefully is resolved.

Overall the data gathering and sequencing methods used in this manuscript are well handled/implemented. The description of research outcomes were well written. Discussion of various methods and approaches in genome sequencing were excellent.